# *Lacticaseibacillus rhamnosus* FM9 and *Limosilactobacillus fermentum* Y57 Are as Effective as Statins at Improving Blood Lipid Profile in High Cholesterol, High-Fat Diet Model in Male Wistar Rats

**DOI:** 10.3390/nu14081654

**Published:** 2022-04-15

**Authors:** Hamza Zafar, Noor ul Ain, Abdulrahman Alshammari, Saeed Alghamdi, Hafsa Raja, Amjad Ali, Abubakar Siddique, Syeda Duaa Tahir, Samina Akbar, Maryum Arif, Metab Alharbi, Abdur Rahman

**Affiliations:** 1Atta ur Rahman School of Applied Biosciences (ASAB), National University of Sciences and Technology (NUST), Islamabad 44000, Pakistan; hz9165661@gmail.com (H.Z.); noorsajid14@gmail.com (N.u.A.); hafsaraja32@gmail.com (H.R.); amjad.ali@asab.nust.edu.pk (A.A.); mabubakar.asab@asab.nust.edu.pk (A.S.); duaatahir98@gmail.com (S.D.T.); 2Department of Pharmacology and Toxicology, College of Pharmacy, King Saud University, P.O. Box 2455, Riyadh 11451, Saudi Arabia; abdalshammari@ksu.edu.sa; 3Department of Pharmacy, Riyadh Security Forces Hospital, Ministry of Interior, P.O. Box 2455, Riyadh 11564, Saudi Arabia; smalghamdi@sfh.med.sa; 4IUT Nancy Brabois, Université de Lorraine, 54601 Villers-lès-Nancy, France; samina.akbar@univ-lorraine.fr; 5Department of Clinical Nutrition, Rawalpindi Institute of Cardiology, Rawalpindi 46000, Pakistan; mariumrajaz@gmail.com

**Keywords:** *Limosilactobacillus fermentum*, *Lacticaseibacllus rhamnosus*, probiotics, cholesterol, obesity, lipid profile

## Abstract

Elevated serum cholesterol is a major risk factor for coronary heart diseases. Some *Lactobacillus* strains with cholesterol-lowering potential have been isolated from artisanal food products. The purpose of this study was to isolate probiotic *Lactobacillus* strains from traditional yoghurt (dahi) and yogurt milk (lassi) and investigate the impact of these strains on the blood lipid profile and anti-obesity effect in a high cholesterol high fat diet model in Wistar rats. Eight candidate probiotic strains were chosen based on in vitro probiotic features and cholesterol reduction ability. By 16S rDNA sequencing, these strains were identified as *Limosilactibacillus fermentum* FM6, *L. fermentum* FM16, *L. fermentum* FM12, *Lacticaseibacillus rhamnosus* FM9, *L. fermentum* Y55, *L. fermentum* Y57, L. *rhamnosus* Y59, and *L. fermentum* Y63. The safety of these strains was investigated by feeding 2 × 10^8^ CFU/mL in saline water for 28 days in a Wistar rat model. No bacterial translocation or any other adverse effects were observed in animals after administration of strains in water, which indicates the safety of strains. The cholesterol-lowering profile of these probiotics was evaluated in male Wistar rats using a high-fat, high-cholesterol diet (HFCD) model. For 30 days, animals were fed probiotic strains in water with 2 × 10^8^ CFU/mL/rat/day, in addition to a high fat, high cholesterol diet. The cholesterol-lowering effects of various probiotic strains were compared to those of statin. All strains showed improvement in total cholesterol, LDL, HDL, triglycerides, and weight gain. Serum cholesterol levels were reduced by 9% and 8% for *L. rhamnosus* FM9 and *L. fermentum* Y57, respectively, compared to 5% for the statin-treated group. HDL levels significantly improved by 46 and 44% for *L. rhamnosus* FM9 and *L. fermentum* Y57, respectively, compared to 46% for the statin-treated group. Compared to the statin-treated group, FM9 and Y57 significantly reduced LDL levels by almost twofold. These findings show that these strains can improve blood lipid profiles as effectively as statins in male Wistar rats. Furthermore, probiotic-fed groups helped weight control in animals on HFCD, indicating the possible anti-obesity potential of these strains. These strains can be used to develop food products and supplements to treat ischemic heart diseases and weight management. Clinical trials, however, are required to validate these findings.

## 1. Introduction

Elevated serum cholesterol levels are a major causal factor in coronary heart disease (CHD). According to the WHO, cardiovascular diseases will remain a leading cause of death and will affect around 23.6 million people worldwide by 2030 [1]. A patient with hypercholesterolemia has a three-fold increased risk of heart attack compared to an individual with normal cholesterol levels [2]. This is linked to the food consumption patterns of people in these countries, who tend to eat high-fat, low-fiber diets. High-fat diets, particularly those high in saturated fatty acids, may raise blood cholesterol levels, increase the risk of atherosclerosis, and cause coronary heart disease [3]. To control blood cholesterol levels, medicines like statins have been used. However, this approach has negative effects, including muscle pain, liver damage, neurological disorders, increased blood sugar, and miscarriages [4]. For this reason, there is a need for developing safer and cost-effective therapies for treating hypercholesterolemia. Probiotics have been proposed as an alternative approach for lowering serum cholesterol levels [5]. The use of probiotics in the development of functional foods is a promising field. Probiotics have been shown to lower blood cholesterol levels by up to 45% [6]. Fermented milk products are an essential part of diets in various parts of world. *Lactobacillus, Lactococcus, Leuconostoc, Pediococcus, Bacillus, Propionibacterium,* and *Bifidobacterium* are among those bacteria which are associated with fermented dairy products [7]. In pre-clinical and clinical studies, milk products have been studied, demonstrating that various lactic acid bacteria lower cholesterol serum levels [8,9]. Traditional fermented foods contain distinct microbial communities that are influenced by artisanal manufacturing techniques and local environmental conditions [10,11]. Furthermore, these findings have demonstrated that artisanal dairy products might be valuable sources for isolating bacterial strains with beneficial probiotic properties [12]. The probiotic *Lacticaseibacillus rhamnosus* CK102 (reclassified from *Lactobacillus rhamnosus*) isolated from fermented yoghurt lowered total cholesterol, high density lipoprotein cholesterol (HDL-c), and triglycerides (TG) by 27.9%, 28.7%, and 61.6%, respectively in a murine model [13,14]. The relation between cardiovascular diseases and total cholesterol levels have been proven by epidemiological studies [15]. Similarly, in another study, *Limosilactobacillus fermentum* MCC2760 (reclassified from *Lactobacillus fermetnum*) lowered cholesterol levels, TG, and low-density lipoprotein cholesterol (LDL-c) levels in a mice model [14,16].

In a rat model, *L. fermentum* PH5 improved blood lipid profiles by lowering cholesterol (67.21%), triglycerides (66.21%), and LDL cholesterol (63.25%). Both *L. fermentum* PH5 and PD2 lowered cholesterol levels in the liver, however *L. fermentum* PH5 performed better than the latter [17].

Although the cholesterol-reducing cholesterol potential of *Lactobacillus spp.* is well known, the comparative data of such strains with clinically used drugs (statin) is less known. This study aimed to investigate the probiotic potential of *Lactobacillus* strains isolated from lassi (butter milk) and dahi (traditional yogurt), particularly their role in shaping the lipid profile in high cholesterol high-fat diet model with Wistar rats. Furthermore, we wanted to compare the impact of these probiotics on serum lipid profile with that of a clinically used drug, i.e., a statin.

## 2. Materials and Methods

### 2.1. Isolation of Lactobacillus from Yogurt and Fermented Milk

Eighty-three samples of artisanal yogurt and traditional fermented milk were collected using standard microbiology procedures from different cities, including Islamabad, Jehlum, Lahore, Khanewal, and Sukkur. After yogurt production, the samples were collected and stored at 4 °C and transported at refrigerator temperature in the Food Microbiology and Biotechnology lab, NUST. The samples were homogenized in PBS and inoculated in MRS broth for partial enrichment. The overnight enriched samples were serially diluted and spread on MRS agar plates supplemented with 1% CaCO_3._ The plates were incubated under anaerobic conditions at 37 °C for 48 h. Colonies were selected based on clear zone formation due to acid production on MRS plates, a property generally attributed to acid producers. Gram-positive rod-shaped and catalase-negative isolates were chosen for further characterization [18].

### 2.2. Gastro Intestinal Tract (GIT) Related Stress

#### 2.2.1. Acid and Bile Tolerance

Isolated lactic acid bacteria (LAB) strains were subjected to gastrointestinal tract (GIT) related stresses. The acid and bile tolerance of the isolates was assessed in 96 well plates [19]. For acid tolerance, 150 µL of MRS broth (HiMedia, Mumbai, India) was adjusted to pH2, and 50 µL of overnight bacterial culture was added to each well. Similarly, MRS broth (HiMedia, Mumbai, India) with 0.3% bile salts (Oxoid, Hampshire, UK) and 50 µL of overnight LAB culture was added to each well for a bile tolerance test. The cell suspensions were incubated anaerobically at 37 °C for 4 h, and growth was measured by absorbance at the 620 nm wavelength. All experiments were performed in triplicate.

#### 2.2.2. Phenol Tolerance

Phenol may be produced by gut microbiota, which can inhibit probiotics, therefore, phenol tolerance by LAB is important for their survival in GIT. Overnight-grown bacterial cultures were inoculated in MRS broth supplemented with 0.4% phenol. Optical density (OD) was measured at 620 nm wavelength in a microplate reader (AMP Platos RII, Graz, Austria) [19].

#### 2.2.3. Lysozyme Tolerance

The lysozyme tolerance test was performed as described previously [20]. The overnight-grown bacterial cultures were harvested by centrifugation at 3500× *g* for 5 min, washed subsequently with phosphate buffer saline (PBS), and re-suspended in PBS. The cell suspension (10 µL) was then transferred into a PBS solution containing lysozyme (20 mg/L) (BioWorld, Visalia, CA, USA). A bacterial culture without lysozyme was used as control. LAB isolates were incubated at 37 °C. OD was measured at 600 nm wavelength after 0 and 90 min to estimate the survival.

### 2.3. In Vitro Safety Assessment

#### 2.3.1. Antibiotic Susceptibility Assay

The disc diffusion method was used to evaluate antibiotic susceptibility of LAB isolates in accordance with clinical and laboratory institute standards. The antibiotics widely used in clinical practice such as Tetracycline 30 µg, Kanamycin 30 µg, Ciprofloxacin 5 µg, Chloramphenicol 30 µg, Gentamicin 10 µg, Amoxicillin/clavulanic acid 30 µg, Streptomycin 30 µg, and Vancomycin 30 µg were used in this study [21].

#### 2.3.2. DNase Activity

Isolates were streaked on DNase agar (Oxoid, Hampshire, UK) and incubated for 48–72 h at 37 °C to observe the clear zone for the DNase activity. *Salmonella enterica* was used as a positive control [9].

#### 2.3.3. Hemolytic Activity

Hemolytic activity of the isolates was assessed using blood base agar containing 7% *v/v* sheep blood. The presence of a pale decolorized halo surrounds the colony after 48 h of incubation at 37 °C was considered positive for hemolytic activity. *Staphylococcus aureus* was used as a positive control [9].

### 2.4. DNA Isolation and Identification

The selected isolates were identified by partial sequencing of the 16S rRNA genes. The DNA extraction was performed as described previously [22]. The extracted DNA was used as a template in PCR for 16S rRNA gene using *Lactobacillus* specific; primers S-17 (AGAGTTTGATTCTGGCTCAG) and A-17 (CAC CGCTACACATGGAG) with the following PCR conditions: 1 cycle of 94 °C for 5 min followed by 35 cycles of 94 °C for 30 s, 52 °C for 45 s, and 72 °C for 30 s, and finally one cycle of 7 min at 72 °C. The purified PCR products of the selected LAB isolates were sequenced. The strains were identified by comparing the sequences with the GenBank database using the Basic Local Alignment Search Tool [23].

### 2.5. In Vitro Cholesterol Assimilating Activity

The capacity of probiotic *Lactobacillus* strains to assimilate cholesterol in MRS broth was investigated as previously described [18]. Cholesterol-PEG 600 (Sigma Aldrich, St. Louis, MI, USA) was added to MRS broth at a final concentration of 100 μg/mL. The 1% (*v/v*) inoculum of each overnight probiotic culture was added and incubated at 37 °C for 24 h. The bacterial cultures were centrifuged at 4000 rpm for 10 min at 4 °C, and the supernatants with non- assimilated cholesterol were collected [24]. Cholesterol assimilation was calculated by the formula:A = (B/C) × 100
where A is the cholesterol that remained with the pellet (as a percentage), B is the absorbance of the sample containing the cells, and C is the absorbance of the sample without bacterial cells. The isolates with the highest cholesterol assimilating potential were selected for in vivo trials. The *Lactobacillus plantarum* ATCC 14917 (KWIK-STIK, UK) was used as a reference strain only for cholesterol assimilation activity.

### 2.6. Bile Salt Hydrolase Activity

The protocol for bile salt hydrolase (BSH) activity measurement was used as described previously with some modifications [13]. Six mm wells were made in MRS agar plates supplemented with 0.3% *w*/*v* bile salts and 0.03% *w*/*v* CaCl_2_. The overnight LAB culture (10 µL) was added and incubated at 37 °C for 72 h in anaerobic conditions. The halos around the wells were observed in order to determine BSH activity. The isolates grown on MRS agar without bile salts were used as the negative control. The experiment was performed in triplicates.

### 2.7. Safety and Survival of Lactobacillus Strains in GIT 2.7.1. Tagging of Isolates

Eight isolates were selected on the basis of in vitro cholesterol assimilation ability and *bsh* activity. All selected isolates were tagged with rifampicin at 200 µg/mL [17].

#### Experimental Groups

Forty-five male Wistar rats aged seven to eight weeks were purchased. These rats were individually housed at 25–30 °C with a 12 h light-dark cycle. The animals were randomly divided into nine groups, with five rats per group. Each group had a similar initial average body weight. The control group was fed with a standard commercial diet and water. The remaining eight groups were provided with the standard commercial diet/normal diet (ND) and 2 × 10^8^ CFU/mL of probiotic dosage in distilled water. For survival assessment, the fecal samples from each group were collected every third day, and the samples were enumerated using the spread plate method. The animals were euthanized after 30 days, and blood samples were collected in a sterile tube. The liver, spleen, small intestine, and large intestine were collected and homogenized in PBS. The homogenized samples were serially diluted, and CFUs were determined by the spread plate method [21].

### 2.8. Hypocholesterolemic Potential of Lactobacillus Strains in the Rat Model

#### 2.8.1. Experimental Groups

Eight strains were selected to evaluate their effect on blood lipid profile. Fifty-five male Wistar male rats aged seven to eight weeks were divided into 11 groups with five animals each. The mean weight of each group ranged between 90 to 110 g. A high-fat high cholesterol diet and normal diet composition are shown (Table 1). The HFCD group was fed with high fat and high cholesterol diet with a feed composition of 79% commercial feed, 10% animal fat, 10% sucrose, and 1% feed grade cholesterol. The SDHF group was fed with a high-fat diet and statin drugs (10 mg/kg body weight), and the ND group was fed with a normal diet (standard commercial feed). The remaining eight groups were provided a high-fat diet with one probiotic strain from each: FM9, Y57, FM6, Y55, FM16, Y59, FM12, and Y63. The probiotic dose of 2 × 10^8^ CFU/mL was administered every day for 30 days in drinking water for each group. The rats were euthanized on the 30th day of dose administration, and a complete anatomical analysis was done.

#### 2.8.2. Serum Lipid Analysis

The total cholesterol, triglycerides, HDL, and LDL levels were measured on the 0 and 30th days of the experiment. Blood was collected by cardiac puncture at 0 day and 30th day. A commercial kit method (Sigma-Aldrich, St. Louis, MO, USA) was used to quantify serum lipids.

### 2.9. Ethical Approval

The ethical approval (IRB-95) for trials was taken from ASAB Institutional Review Board on 6 December 2018.

### 2.10. Statistical Analysis

All the data were expressed as the standard error of mean ± SEM. Significance was measured by performing an ANOVA followed by Bonferroni’s post-hoc test. Data were analyzed using GraphPad Prism (version 5.01) (GraphPad Software, San Diego, CA, USA). Significance was accepted at *p* < 0.05 and *p* < 0.001.

## 3. Results

### 3.1. In Vitro GIT Stress Tolerance and Safety Assessment

The survival of isolates in the GIT is an essential feature of probiotics. All the selected isolates were tolerant to GIT related stress, including acid, bile, phenol, and lysozyme stress (Appendix A. The tolerance to such gastric stress indicated their potential to survive in GIT, a key attribute for probiotics. All the selected strains, FM9, Y57, FM6, Y55, FM16, Y59, FM12, and Y63, survived in broth with pH2 and 0.3% bile for 4 h. All the LAB isolates were subsequently evaluated for their phenol resistance, where these showed increased tolerance to 0.4% phenol with OD values >1.000 after 24 h of incubation. All the tested LAB isolates showed significant resistance to lysozyme (20 mg/L) after 90 min (Appendix A).

### 3.2. Safety Assessment

#### DNase, Hemolytic, and Antibiotic Susceptibility Assay

All 8 selected LAB isolates showed negative hemolytic (γ-hemolysis) results and DNase activity. However, a variable resistance was observed for different tested antibiotics. No zone was observed against Gentamycin, Vancomycin, Kanamycin, and Streptomycin, demonstrating complete resistance to these antimicrobials in these LAB isolates (Table 2). Most of the isolates were susceptible to tetracycline and amoxycillin/clavulanic acid.

### 3.3. Molecular Identification of LAB Strains

By comparing the 16S rRNA nucleotide sequences of strains, LAB isolates were identified as *Limosilactobacillus fermentum* FM6, *Lacticaseibacillus rhamnosus* FM9, *L. fermentum* FM12, *L. fermentum* FM16, *L. fermentum* Y55, *L. fermentum* Y57, *L. rhamnosus* Y59, and *L. fermentum* Y63 with the GenBank accession numbers, MW521133, MW556565, MW556566, MW521133, MW566788, MW563932, MW563933, MW563934, respectively.

### 3.4. In Vitro Cholesterol Assimilation

Selected *Lactobacillus* strains isolated from traditional yogurt and buttermilk were evaluated for their cholesterol-lowering capability. A variable (37–49%) cholesterol assimilation was found in the *Lactobacillus strains* (Figure 1). Among nine strains, positive control *L. plantarum* ATCC 14917 exhibited the highest cholesterol-lowering potential (49%), followed by FM6 (43.19%), FM9 (41%), and Y59 (41%) (Figure 1).

### 3.5. In Vivo Survival and Safety Assessment

#### Survival and Colonization Preference in GIT

The rifampicin-tagged isolates were recovered in the fecal sample, indicating that they survived in the GIT. The bacterial count gradually increased from the first trial and remained stable on the 19th day (Figure 2). However, no bacteria were retrieved from the negative control group, indicating that no cross-contamination occurred between groups during the trial.

All strains showed colonization in both the small and large intestines; however, strains showed better colonization in the large intestine than in the small intestine. Maximum colonization obtained by *L.*
*fermentum* FM6 and *L. rhamnosus* Y59 were 2.55 × 10^7^ cfu/g and 2.72 × 10^7^ cfu/g, respectively, in the large intestine. No colonization was observed in the control group, blood, liver, or spleen samples from all groups, indicating that bacteria were not translocated from the gut to these organs, highlighting the potential safety of these strains (Figure 3).

### 3.6. Total Lipid Profile Analysis in the Rat Model

#### 3.6.1. Cholesterol Level

On day 0, no significant difference was observed in serum cholesterol levels between the seven groups. However, the HFCD group significantly increased (31%) in cholesterol levels at day 30, whereas all probiotic groups had a moderate decrease in blood cholesterol levels. *L. rhamnosus* FM9 reduced serum cholesterol by 9%, while *L. fermentum* Y57 and *L. fermentum* FM6 reduced cholesterol by 8% and 7%, respectively, which was significantly better than the SDHF group, where the values were 5% (Figure 4).

#### 3.6.2. HDL Level

HDL levels in the HFCD group significantly decreased by 8%. The HDL level in the SDHF group, *L. rhamnosus* FM9, *L. fermentum* Y57, *L. fermentum* FM6, *L. rhamnosus* Y55, *L. fermentum* Y59, and *L. fermentum* FM16 was significantly elevated by 46%, 46%, 44%, and 44%, 37%, 22%, and 16% respectively. The data reveals that increased HDL is equally good in *L. rhamnosus* FM9 and *L. fermentum* Y57 as by statin-treated group (Figure 5).

#### 3.6.3. LDL Level

The LDL significantly increased by 33% in the HFCD group, while all probiotics could decrease the LDL levels. However, a highly significant decrease was observed in the case of *L. fermentum* Y57, *L. rhamnosus* FM9, and *L. fermentum* FM6, with 41%, 37%, and 31% values, respectively. At the same time, statin reduced LDL levels to only 21%. This shows that the impact of *L. fermentum Y57* and *L. rhamnosus FM9* was much better in modulating LDL levels than atorvastatin (Figure 6).

#### 3.6.4. Triglyceride Level

At day 30, there was no significant difference between the probiotic treatment groups and the statin treatment groups. However, when compared to the other groups, the HFCD group’s triglyceride level increase was extremely significant when assessed using Bonferroni’s post hoc analysis (Figure 7).

#### 3.6.5. Impact of Isolates on the Average Weight of Animals in Normal Feed Regime

Weight gain is used as a standard for animal health. There was a non-significant difference in body weight between the groups at day 0; however, a healthy weight gain was observed in all test groups, including probiotic-fed groups. The percent weight gain varied in different groups, where probiotic fed groups demonstrated lower weights compared to the control group (group on normal diet), after 30 days (Figure 8A). At day 30, groups fed *L. rhamnosus* FM9, *L. fermentum* FM12, *L. fermentum* FM6, and *L. rhamnosus* Y59 strains gained considerably less weight than the control group (Figure 8B). These data indicate that these strains have a considerable impact on weight control and may have anti-obesity properties. These findings indicate that the strains may have an anti-obesity effect.

#### 3.6.6. Impact of Strains on Weight in High Fat and Cholesterol Feed Regime

Weight gain was relatively modest in the test groups compared with the HFCD group. However, there was a significant difference in all probiotic groups compared to the control groups (ND: normal diet, SDHF: high-fat diet, while treating with statins, and HFCD: high-fat diet with no treatment) at day 30 (Figure 8B). These findings reveal that these strains, *L. rhamnosus* FM9 and *L. rhamnosus* Y59, also showed potential to act as a weight watcher, as it moderated the weight gain compared to the control groups (ND, HFCD, and SDHF).

When feed intake was estimated for each group, it was observed that there was no significant difference in feed intake between the groups during both trials. This indicates that weight moderation is not due to lesser feed intake.

## 4. Discussion

Probiotics have demonstrated the ability to lower serum triglycerides, cholesterol, and low-density lipoprotein. As a result of these findings, there is growing interest in employing probiotics to prevent coronary heart disease [2]. The strains FM9, Y57, FM6, Y55, FM16, Y59, FM12, and Y63 were recovered from traditional fermented milk (lassi) and artisanal yogurt in this study. These isolates demonstrated in vitro tolerance to gastrointestinal tract-related stresses (acid, bile, lysozyme, phenol) and bile salt hydrolase potential. Probiotics need to withstand gastrointestinal stress conditions, such as lysozyme, phenol, acidity, and bile, to exert a beneficial role in GIT. Our findings were consistent with a prior work in which the *L. fermentum* AD1 strain isolated from fermented milk demonstrated good survival at low pH, 0.3% bile salt, and 0.4% phenol isolated from fermented milk [25].

Due to their long history of safe use in foods, most probiotic strains are GRAS (Generally Recognized as Safe) [26]. Nevertheless, researchers identified some LAB strains as opportunistic pathogens capable of infections in humans. As a result, establishing the safety of these strains is essential before using them as probiotics [27]. In our study, all of the initially screened isolates exhibited negative results for hemolytic, DNase activity, and susceptibility to the many antibiotics tested. These findings are consistent with recent studies in which *L. fermentum* NCU3087 and *L. fermentum* NCU3088 exhibited no hemolysis or DNase activity [28]. The isolated LAB should be antibiotic sensitive to avoid the transfer of undesirable antibiotic resistance to the intestinal microbiota [29]. According to previous findings, *L. rhamnosus* and *L. fermentum* were susceptible to methicillin, penicillin, tetracycline, levofloxacin, azithromycin, chloramphenicol, and other antibiotics [30]. Based on 16S rRNA gene sequence analysis, the isolates FM6, FM12, FM16, Y55, Y57, and Y63 were 99–100 percent similar to *L. fermentum*, while FM9 and Y59 were 100 percent similar to *L. rhamnosus*. Previous research has also found *L. fermentum* and *L. rhamnosus* in fermented milk [31,32].

Many in vitro tests are used to select cholesterol-lowering probiotics, including assessing bile salt hydrolase (BSH) activity. Prior studies [33] showed that *Lactobacillus* strains with Bile salt hydrolase activity reduced cholesterol more effectively when administered in a rat model. The in vitro cholesterol-lowering activity was variable in our isolates, ranging from 19% to 49%. The highest cholesterol-lowering potential was observed in the FM6 strain (38%), followed by Y59 (36%). In a prior study, *L. rhamnosus* BFE5264 demonstrated in vitro cholesterol-lowering potential, suggesting that it could help decrease plasma total cholesterol concentrations [34]. However, a weak cholesterol reduction potential (5%) was observed in *L. plantarum* [35].

The selected strains met the criteria for being identified as a potential probiotic strain and subsequently were used to investigate their probiotic effects in rats fed a high-cholesterol diet. Probiotics exert physiological functions only when they effectively survive in the host GIT and propagate through colonization [36]. The *Lactobacillus* strains in this study showed the ability to resist GIT stress when tested in the rat model. The bacterial count in the feces was low during the first week of the study. Nonetheless, the bacterial count in the feces continues to rise, indicating that these strains could survive and colonize in the GIT. Similar findings have been reported in previous studies [37] where different probiotic strains effectively colonize in the GIT of the rat. In our study, colonization was seen in both small and large intestines; however, greater colonization of strains was observed in the large intestine, indicating that these strains preferred the anaerobic environment of the large intestine. Consistent with our findings, a previous study [38] discovered that probiotic *Lactobacillus* strains prefer the large intestine. According to an earlier study, the colonization and distribution of *L. plantarum* MA2 were observed in the small intestine of mice [39]. The colonization of LAB in the gut play an important role in their beneficial effects on the host.

Another study found that *L. rhamnosus* SKG34 prefers colonization in the large intestine [40]. There was no evidence of bacterial translocation in the liver, blood, or spleen, indicating that our strains are nonpathogenic. These findings are consistent with prior research, which found that *L. fermentum* SM-7 did not cause translocation in the blood, liver, or kidneys and was safe to use [41]. In another study, *L. fermentum* PL9005 did not translocate from the gut to other organs in mice [28].

The cholesterol-lowering ability of selected strains was validated in a rat model. This study revealed that *L. rhamnosus* FM9, *L. fermentum* Y57, and *L. fermentum* FM6 increased serum HDL levels by 46%, 46%, and 44%, respectively. These findings were comparable to those reported previously, where administration of *L. rhamnosus* and *L. fermentum* elevated HDL-c levels by 15.23% and 20.12%, respectively [42]. In another study, no significant effect was observed by the *L. fermentum* strains isolated from human feces on HDL in a mice model [43]. This impact may be strain-dependent and a source of isolation [31]. In our study, the supplementation of *L. fermentum* Y57, *L. rhamnosus* FM9, and *L. fermentum* FM6 showed a significant decrease in serum LDL by 41%, 37%, and 31%, respectively. According to a previous study, *L. fermentum* strains showed a significant decline in LDL levels, at 51.07%, 49.73%, 57.96%, and 63.25%, respectively. [14]. According to [42], decreased levels of cholesterol and increased levels of HDL content may reduce the conversion of intermediate-density lipoprotein to LDL-C particles, resulting in a decrease in serum LDL-C concentration. This could be the reason for the probiotics’ LDL-C lowering effect. However, in previous reports, probiotic strains did not affect LDL levels in a rat model [44]. There was no significant difference in triglyceride levels observed on day 30 in probiotic treatment groups and statins, while the level of triglycerides was significantly high in the HFCD group. These results were in agreement with the previous study where *Lactobacillus* strains have no impact on triglyceride levels in rats after a four week trial [45]. However, according to a previous study, the administration of *L. fermentum* strains decreases triglycerides by 66.3%, 65.3%, 50.5%, 70.7%, and 66.2%, respectively [17]. According to Harisa et al., [46] the hypotriglyceridemic effect of probiotics may be attributed to the activation of lipases, a decrease in intestinal lipid absorption, or an increase in lipid catabolism and/or antioxidant activity. Lipoprotein lipase is responsible for TG metabolism, which results in normalization of its plasma level, while the increase in triglyceride levels could potentially be attributed to high fat stimulation of hepatic very low density lipoprotein (VLDL) synthesis [47]. A previous study also reported that *L. rhamnosus* had no impact on triglycerides [40]. In an earlier study, *L. fermentum* SM-7 was isolated from a fermented milk drink (koumiss), and it significantly reduced total serum cholesterol (TC) and LDL. However, it did not increase HDL-C significantly in the mice model [48]. The mechanism by which *Lactobacillus* strains lower cholesterol levels and improve lipid profiles remains unknown. However, this is presumed due to the probiotics’ ability to modulate rats’ gut microbiota’s growth and fermentative products [16]. There are four well-documented mechanisms by which probiotics lower cholesterol levels in the blood. These include (1) the probiotic bacteria’s ability to assimilate cholesterol molecules in the small intestine; (2) the ability of probiotic microbes to enzymatically deconjugate bile acid using bile salt hydrolase (BSH). Because bile acid is mostly dissolved in the conjugated form, only a small portion is absorbed in the intestine, and the rest is excreted in the feces. The absorbed cholesterol is then used to synthesize new bile acids (a homeostatic response), resulting in a decrease in serum cholesterol level; (3) the conversion of cholesterol to coprostanol by cholesterol reductase of lactobacillus strains; and (4) products of the *Lactobacilli* fermentation process in the form of short chain fatty acids [38].

The administration of *Lactobacillus* strains FM6, FM9, FM12, Y55, and Y59 in the HFCD mice significantly decreased the weight gain compared to the control group in a 30-day trial, which indicates that it reduces fat deposition and obesity. Similar findings were observed in a previous study where *Lactobacillus* strains significantly reduced body weight in a rat model [49]. In another study, some of the *Lactobacillus* strains do not affect the bodyweight of the mice [50]. According to a previous study, the administration of *L. fermentum* CRL1446 resulted in a significant reduction in body weight in mouse models, reaching values similar to the control [51]. The selected probiotic strains in this study revealed the potential to improve serum lipid profile and obesity control. However, two strains, *L. rhmanosus* FM6 and *L. fermentum* Y59, prove to be the best candidates in reducing total cholesterol, triglycerides, LDL, and improving HDL since their performance is as good as the statins treated group. Probiotic consumption also improved weight control in animals on a high-fat, high cholesterol diet. Although the potential of *Lactobacillus* in improving lipid profiles is well known, however, its comparison with clinically used drugs is not well known. This study suggests that *L. rhamnosus* FM9 and *L. fermentum* Y57 strains exhibit great potential in improving the lipid profile, which is as good as statin.

## 5. Conclusions

This study demonstrates that *L. rhmanosus* FM9 and *L. fermentum* Y57 best reduce hypercholesterolemia among all tested probiotic strains. These strains’ impact on cholesterol reduction and overall lipid profile was equally as good as statins in Wistar rats fed high fat, high cholesterol diets after 30 days. These strains also improved weight control and may have anti-obesity potential. Our study also has some limitations: we used rats, and the observed effects in an animal model may not be the same in humans. Therefore, clinical trials need to be conducted to validate the efficacy and safety of the strain for its use in managing high cholesterol in humans. Furthermore, these strains can be a source of an adjunct or a starter culture in different food products such as fermented milk, yogurt, and other dairy foods.

## Figures and Tables

**Figure 1 nutrients-14-01654-f001:**
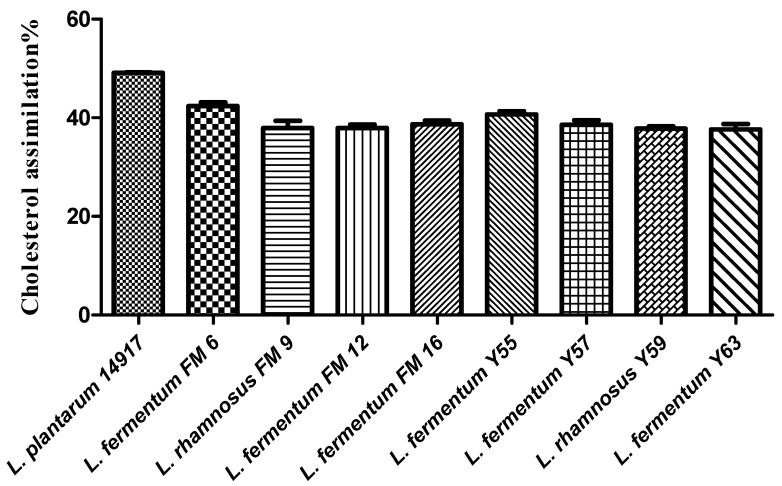
Cholesterol assimilation (%) of *isolates* in MRS broth supplemented with bile.

**Figure 2 nutrients-14-01654-f002:**
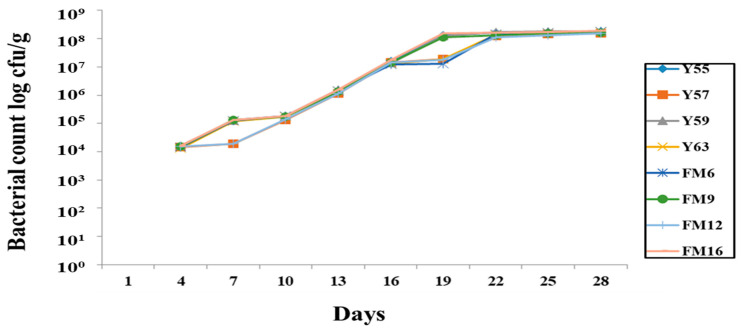
Recovery of rifampicin-tagged LAB isolates from male Wistar rat feces.

**Figure 3 nutrients-14-01654-f003:**
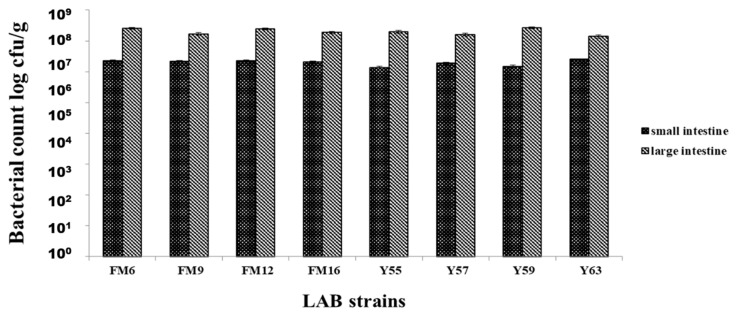
Adhesion of LAB strains to small & large intestines of rats after 30 days.

**Figure 4 nutrients-14-01654-f004:**
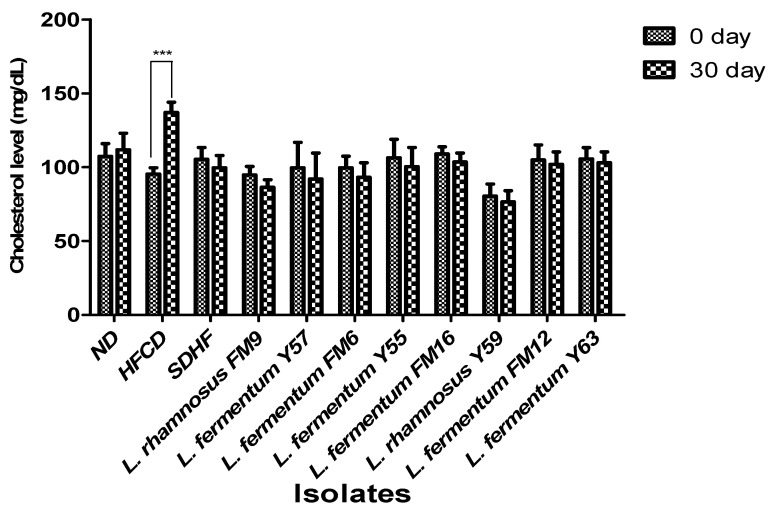
Impact of probiotic feeding on total serum cholesterol levels in experimental groups (Intragroup at day 0 and 30). *** *p* < 0.001.

**Figure 5 nutrients-14-01654-f005:**
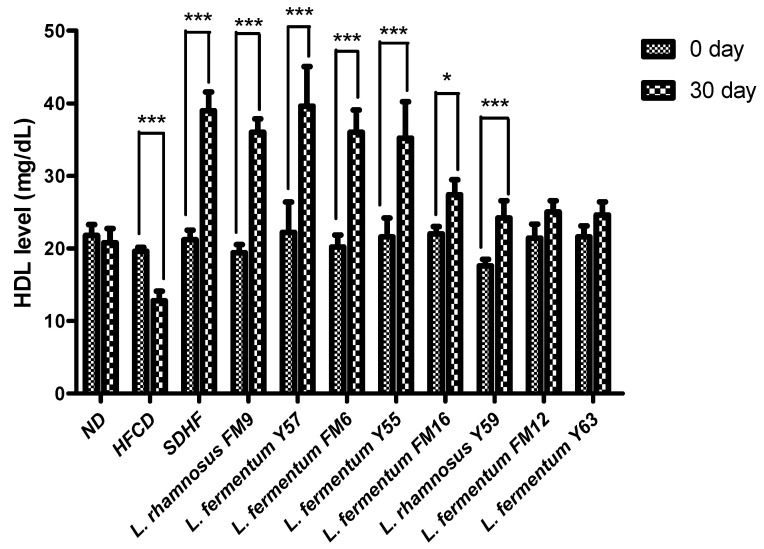
Intra group comparison of HDL level at day 0 and day 30. * *p* < 0.05, *** *p* < 0.001.

**Figure 6 nutrients-14-01654-f006:**
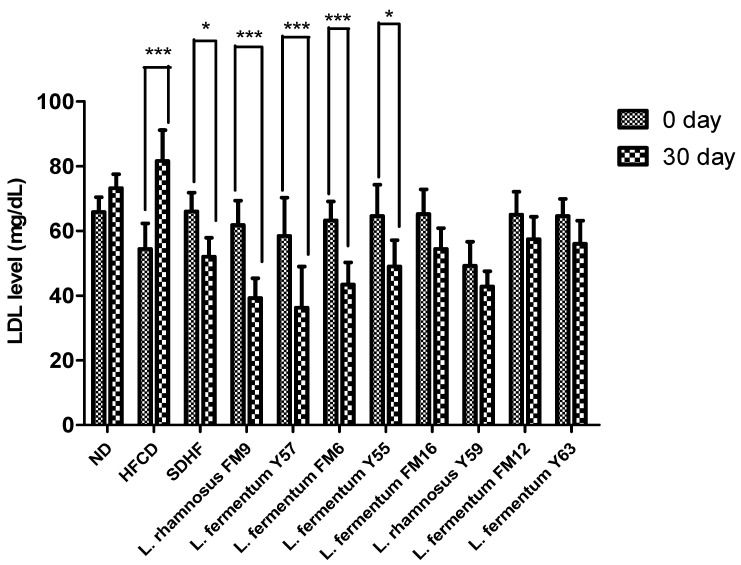
Intra group comparison of LDL level at day 0 and day 30. * *p* < 0.05, *** *p* < 0.001.

**Figure 7 nutrients-14-01654-f007:**
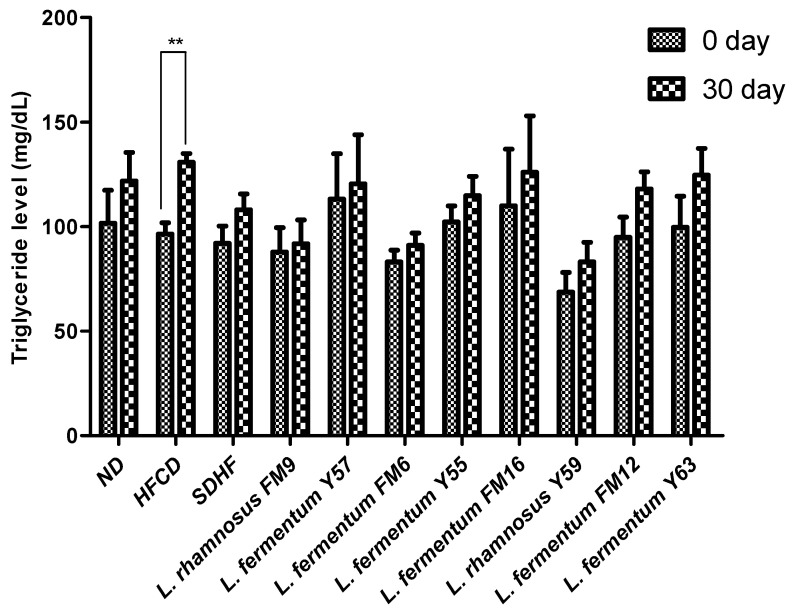
Intra group comparison of Triglyceride level at day 0 and day 30. ** *p* < 0.01.

**Figure 8 nutrients-14-01654-f008:**
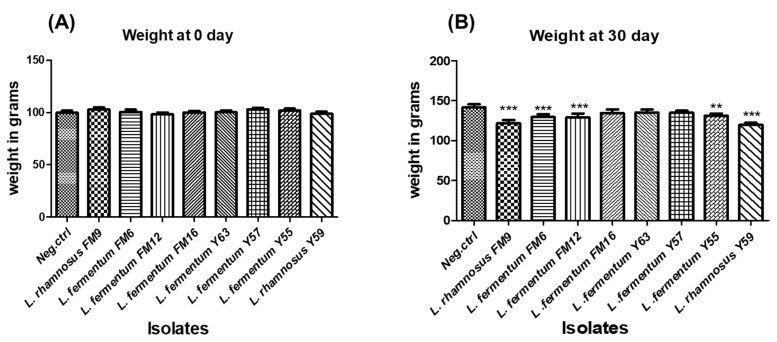
The average weight of rats administered with different LAB at (**A**) day 0 and (**B**) day 30. ** *p* < 0.01, *** *p* < 0.001.

**Table 1 nutrients-14-01654-t001:** Composition of diets used in the study.

Ingredients	High Cholesterol High Fat Diet (HFCD) (%)	Normal Diet (ND) (%)
Carbohydrate	48	48
Protein	19	24
Fat	13	4
Moisture	10	12
Fiber	3	4
Ash	6	8
Cholesterol	1	0

**Table 2 nutrients-14-01654-t002:** Antibiotic susceptibility profile of LAB isolates.

Isolates	CIP	CN	K	VA	C	SR	RD	TE	AMC
**FM6**	S	R	R	R	S	R	S	S	S
**FM9**	S	R	R	R	S	R	S	S	S
**FM12**	S	R	R	R	S	R	S	S	S
**FM16**	R	R	R	R	R	R	S	S	S
**Y55**	S	R	R	R	R	R	S	S	S
**Y57**	S	R	R	R	R	R	S	I	S
**Y59**	S	R	R	R	R	R	S	I	S
**Y63**	S	R	R	R	S	R	S	I	S

Abbreviations: C: chloramphenicol (30 μg); AMC: amoxicillin-clavulanic acid (10 μg); CIP: ciprofloxacin (10 μg); CN: gentamicin (10 μg) K: kanamycin (30 μg; VA: vancomycin (30 μg); RIF: rifampicin (30 μg); TE: tetracycline (30 μg); SR: streptomycin (5 μg). R: resistance; S: sensitivity; I: intermediate.

## Data Availability

Not applicable.

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
