# Peer review of "Lacticaseibacillus rhamnosus* FM9 and *Limosilactobacillus fermentum* Y57 Are as Effective as Statins at Improving Blood Lipid Profile in High Cholesterol, High-Fat Diet Model in Male Wistar Rats"

_nutrients, 2022, doi:10.3390/nu14081654_

Round 1

Reviewer 1 Report

The manuscript presents an interesting topic connected with cholesterol-lowering potential by probiotic products. Nevertheless, as a reviewer, I must indicate numerous deficiencies in the text. Please explain doubtfully, listed below aspects.

Abstract

Line 21: Lactobacillus word should be written in italics

Line 21: The purpose of this study was not only to „isolate probiotic Lactobacillus strains from traditional yogurt (Dahi) and yogurt milk (lassi)” but also to investigate the impact of isolated strains on blood lipid profile

Line 22: „that can improve... obesity...” - this statement requires a change

Line 22: „Dahi” - is a capital letter required?

Line 26: „The safety was determined by giving Wister rats an oral suspension.” - please explain this unclear sentence

Lines 26 and 29: Wister word should be changed on Wistar

Line 34-37: Please add information if these changes were statistically significant

Line 35: a „percent” word should be changed on „%”

Line 37: „These strains gained significantly less weight” - this statement requires a change

Introduction

Line 50:The WHO document should be quoted at this point, not research of Nag, T., and Ghosh, A. (2013)

Line 51: What types of dietary supplements are commonly used? Some examples of nutritional supplements should be replaced based on scientific reports.

Line 67: The sentence refers to: „Numerous studies indicate..”, while only one scientific research was quoted (Koirala et al. 2013).

Line 80: „Model group” - is a capital letter required?

Material and methods

Line 94:What were the conditions for transporting and storing samples? How much time has passed since yogurt production to the stage of samples collection?

Line 101: Abbreviations: LAB, GIT, MRS, CLSI, BSH, PBS, etc. should be explained at the first use in the text

Line 104: The order of citing position from the bibliography was not preserved.

Line 106: „MRS broth (HIMEDIA, UK) with 0.3% bile salts (oxbile) and 50 µL of overnight LAB culture for bile tolerance.” - unfinished sentence.

Line 157: The „Ethical approval” section should be directly above "Statistical Analysis"

Line 157: Ethical issues require clarification. Did the animal ethics committee approve the study protocol? What was the date of obtaining consent? Were the research team members skilled in the conduct of animal research?

Line 168: Two more groups should be extracted: fed HFCD and SDHF diets.

Line 172: „The rats were euthanized after 28 days, and blood samples were.” - unfinished sentence.

Line 172: Please specify whether: „The rats were euthanized after 28 days” or „The rats were euthanized on the 30th day” (line 186)?

Line 180: Table 1 shows the antibiotic susceptibility profile of LAB isolates, not a diet composition.

Line 182: How much was the dose of statins in a feed?

Lines 180-183: A detailed HFCD, SDHF, and ND diets composition are required.

Line 194: Statistical analysis section require clarification. What was the significance level assumed? Was a normal distribution demonstrated in each group (using what statistical test?)? How was the homogeneity of variance tested in each group? Why was Bonferroni’s test chosen for analysis?

Results

Lines 202-201: Abbreviations for streptomycin (S) and sensitivity (S) should be different between each other

Line 231: From the beginning, only eight strains were mentioned in the text (without L. plantarum ATCC 14917). Where did the 9th strain be obtained from?

Line 557: Which significantly different cholesterol levels characterized groups on day 0?

Line 266: a „percent” word should be changed on „%”

Line 269: Why did the authors not mention L. fermentum FM6, L. fermentum Y55, L. rhamnosus Y59, and L. fermentum FM16, for which statistical significance was also demonstrated?

Line 273: a „percent” word should be changed on „%”

Line 296: “weight watchers” is a colloquial expression

Line 297: The results in this section should also be presented as numerical values

Line 301: Figure 9b is not included in the manuscript

Line 310: The *** symbols should be explained below the figure

Discussion

Line 314: citation is required

Line 332: Table 2 is not included in the manuscript

Line 318: „Lassi” - is a capital letter required?

Line 345: “Bile” - a capital letter is not required

Line 347: “In our study, L. Plantarum ATCC 14917” This strain is not mentioned in the abstract, introduction, or material and methods section

Lines 372-374: L. fermentum Y57, L. rhamnosus FM9 372 and L. fermentum FM6 reduced TGs by 41%, 37%, and 31%, respectively which are significantly better than statin treated group.” As shown in Figure 7 all probiotic strains caused an increase in triglycerides level, which proves that the authors drew incorrect conclusions.

Line 376: “and 66.2” missing “%”

Line 383: citation is required

Line 390: Lactobacillus word should be written in italics

Morover:

The discussion is very general, and few probiotic genera and strains are quoted, suggesting to readers that generalization of obtained results might be possible. The discussion is one-sided. The authors do not cite any counter-arguments.

The results are presented in an unclear way. There are discrepancies in the basic assumptions of the study (e.g., study duration 28 or 30 days? Number of strains tested 8 or 9?). The authors indicate that the probiotic strains increased the blood TG level (line 282) and then suggest that tested strains decreased this level (line 373).

Extensive editing of English language and style required. There are many grammatical errors, colloquial phrases, and repetitions in the manuscript. The writing style does not correspond to the scientific style.

The bibliography is poor (only 36 items), the style of references does not meet the Nutrients requirements.

Sections: Author Contributions, Funding, Institutional Review, Board Statement, Informed Consent Statement, Data Availability Statement, Conflicts of Interest have not been completed.

Although the study's subject and design are interesting and have application potential, the way of presenting the results, numerous mistakes, and inaccuracies mean that, in my opinion, this manuscript does not deserve to be published in the Nutrients.

Author Response

Dear Reviewer, Please find the response attached. thanks

Reviewer 2 Report

The Manuscript entrusted to me for review focuses on a significant subject. The research concerns the properties of potentially probiotic Lactobacillus strains isolated from food products in the context of reducing blood cholesterol levels. Although the topic is interesting and justified due to the increase in coronary diseases caused by an abnormal lipid profile, I have some comments to the Authors of the work.
First of all, the Manuscript requires linguistic correction. In places the language is so chaotic that it is hard to understand what the Authors meant. Grammar is limited.
Moreover, a new taxonomic classification for LAB was introduced in 2020
(eg. https://www.microbiologyresearch.org/content/journal/ijsem/10.1099/ijsem.0.004107 – new reclassification of the genus Lactobacillus based on the whole genome sequencing of Lactobacillaceae and Leuconostocaceae). In my opinion, new and valuable scientific publications should be based on the latest nomenclature, consistent with the latest scientific reports. Therefore, nomenclature used in this Manuscript is out of date and unacceptable. Keeping in mind that this new classification is “just fresh” it is good to present two names, according to old and new nomenclature to emphasize that the Authors are up to date with the latest scientific achievements/reports.
The Authors in the Abstract wrote: "Lactobacillus spp., a probiotic that can lower plasma cholesterol levels". Such a sentence is unacceptable. Not every representative of Lactobacillus is a probiotic, and what's more - the properties of probiotics are strain-dependent, so assigning probiotic traits and the properties of lowering plasma cholesterol levels for all representants of Lactobacillus genera is a major abuse.
“The purpose of this study was to isolate…” – the aim of the study cannot be “isolation”, or at least it seems to me that this was not the purpose of the Authors’ work. It seems to me that the aim of the Authors’ work was to assess the probiotic properties of potentially probiotic Lactobacillus strains isolated from... and in particular to assess their role in shaping the lipid profile in...
The abstract, especially the second part, is very chaotic, it is hard to understand what the study was about
Line 55: “technique” – that’s not a good word
The Introduction section is written in very heavy English. It's hard to read... I consider Paragraphs 75 to 84 redundant. It describes the previous study, not just the previous findings as is usually the case in the Introduction.
Line 85-86: The Authors should to write it differently! In this form, it is unknown what is the legitimacy of the Authors’ research! – comparison? It makes sense, but needs to be redrafted. Next – please write the purpose of the research correctly!
Line 93-94: Microbiological procedures came from different cities?
Line 101: Do not use abbreviations in the names of headings / subheadings. There is no explanation for GIT in the Manuscript. I can understand this is a gastrointestinal tract, but not everyone has to understand it. Abbreviations need to be explained.
Line 105: pH=2
Line 105-106: “MRS broth (HIMEDIA, UK)” – producer, city, country – given when used for the first time. Such information is given to all the most important chemicals/reagents. In this Manuscript it is lacking
Line 108: wavelength?
The various sections of Materials and Methods have a good idea, but are described inconsistently with the art. The lysozyme section is best described. Please improve.
Line 165: Wistar
Line 172: blood samples were what?
Figure 1. What is the meaning of different labeling for different strains?
I have no major objections to the graphical presentation of the results. Perhaps I will only suggest that several Figures are analogous and could be panned/connected
Line 396: with probiotic with potential probiotic strains ???
In conclusion, I believe that the Manuscript has a very great potential. The concept of individual sections is correct, but the descriptions do not comply with the publication standards. I have the impression that all the shortcomings are due to poor English. I suggest improving the Manuscript, especially linguistic proofreading and sending it after responding to the review.

Author Response

Dear Reviewer,

Thank you for your review. Please find attached the detailed response. thanks

Reviewer 3 Report

The paper has an useful topic, with immportant applications. 

The article is well written and neat. Small leaks.

Line 21: please use italic for Lactobacillus

Please use either % or "percent" (i.e. line 35).

The subtitles, please write them according to the journal requirements. The author are using randomized cappital or small letters. 

2.3. In vitro

Author Response

Dear Reviewer,

Thank you for your comments. Please find our detailed response. thanks

Round 2

Reviewer 1 Report

Current manuscript is definitely better than the previous version. A few comments are placed below:

Line 22:

I suggest: The purpose of this study was to isolate probiotic Lactobacillus strains from traditional yohurt (dahi) and yogurt milk (lassi) and investigate the impact of these strains on blood lipid profile and anti-obesity effect in a high cholesterol high fat diet model in Wistar rats

Line 53:

 „person” word should be changed on „patients”

Line 227:

The assumptions of Bonferroni test indicate that all variances are equal, and all averages are characterized by standard distribution. Have authors checked these assumptions? If yes, which tests have been used? If the assumptions have not been met, another test should be used (C Dunnetta or Tamhane's T2).

Line 314:

 What could be the reason for the growth of triglycerides levels in all groups? Perhaps it is worth referring to possible reasons in the discussion...

General

English and style still require some changes. I suggest to use MDPI or native speaker language correction.

Author Response

Dear Reviewer, 

Thank you for your worthy comments. We have addressed the comments and the detailed response is attached. Thanks

Best Regards,

Abdur Rahman

Reviewer 2 Report

The aim of the research is still poorly worded, even in the Abstract. The isolation of strains cannot be the aim of the research. Why did the Authors write "in vitro" and not "In vitro" in the headings of individual sections? When the first word of the title is "In vitro", in the text "in vitro". I can see the authors' great commitment to improving the Manuscript, but the work has "technical" shortcomings regarding the standards of writing scientific publications. 

It seems to me that the Manuscript needs to be given a greater "flow" and the writing style to be more scientific, typical of scientific publications.

Author Response

Dear Reviewer,

Thank you so much for providing us the opportunity to improve the manuscript. We are thankful for your worthy comments and suggestions. We have addressed these comments, the detailed response is attached. Thanks

Best Regards,

Abdur Rahman
